# Attenuation of Tumor Development in Mammary Carcinoma Rats by Theacrine, an Antagonist of Adenosine 2A Receptor

**DOI:** 10.3390/molecules26247455

**Published:** 2021-12-09

**Authors:** Cian-Fen Jhuo, Yu-Yu Hsu, Wen-Ying Chen, Jason T. C. Tzen

**Affiliations:** 1Graduate Institute of Biotechnology, National Chung-Hsing University, Taichung 402, Taiwan; quartz1248s@gmail.com (C.-F.J.); 410413012@gms.ndhu.edu.tw (Y.-Y.H.); 2Department of Veterinary Medicine, National Chung-Hsing University, Taichung 402, Taiwan

**Keywords:** adenosine 2A receptor, anti-tumor immunity, mammary carcinoma, theacrine

## Abstract

Caffeine has been reported to induce anti-tumor immunity for attenuating breast cancer by blocking the adenosine 2A receptor. Molecular modeling showed that theacrine, a purine alkaloid structurally similar to caffeine, might be an antagonist of the adenosine 2A receptor equivalent to or more effective than caffeine. Theacrine was further demonstrated to be an effective antagonist of the adenosine 2A receptor as its concurrent supplementation significantly reduced the elevation of AMPK phosphorylation level in MCF-7 human breast cells induced by CGS21680, an agonist of adenosine 2A receptors. In an animal model, the development of mammary carcinoma induced by 7,12-Dimethylbenz[a]anthracene in Sprague–Dawley rats could be attenuated by daily supplement of theacrine of 50 or 100 mg/kg body weight. Both expression levels of cleaved-caspase-3/pro-caspase-3 and granzyme B in tumor tissues were significantly elevated when theacrine was supplemented, indicating the induction of programmed cell death in tumor cells might be involved in the attenuation of mammary carcinoma. Similar to the caffeine, significant elevation of interferon-γ and tumor necrosis factor-α was observed in the serum and tumor tissues of rats after the theacrine supplement of 50 mg/kg body weight. Taken together, theacrine is an effective antagonist of adenosine 2A receptors and possesses great potential to be used to attenuate breast cancer.

## 1. Introduction

Breast cancer is a threatening neoplasm of the female population, and its incidence rate has increased continually all over the world over the past few decades [1]. Surgery is commonly the first treatment for breast cancer, followed by other supporting systemic remedies, including chemotherapy, radiotherapy, hormone therapy, and specifically targeted drug therapy [2]. However, the therapeutic effects of these treatments in some cases are not satisfactory, and severe side effects are usually observed in most patients [3]. There is a growing demand to develop new medical therapies for breast cancer treatment with better therapeutic effects and less adverse effects.

Immunotherapy is currently considered an alternative remedy for breast cancer patients as it stimulates patients’ own immune systems to effectively recognize and destroy cancer cells [4]. Several drugs have been approved to be used in the immunotherapy of breast cancer, such as immune checkpoint blockades, PD-L1 and CTLA-4 [5,6]. Recently, regulation of the adenosine 2A receptor was proposed to be a prospective target for immunotherapy, and antagonists of the adenosine 2A receptor were demonstrated to be potential drugs to enhance the immunological response for the attenuation of the development of mammary carcinoma [7,8,9,10].

Caffeine and theophylline have been reported to induce an anti-tumor immune response for attenuating breast cancer by blocking the adenosine 2A receptor [11,12]. Theacrine, a purine alkaloid structurally similar to caffeine, is more bitter than caffeine in taste and is a relatively abundant constituent in Yunnan Kucha tea [13]. Although theacrine and caffeine possess a similar structural backbone, they display quite different physiological functions in the central nervous system. In a previous study, pretreatment of theacrine prolonged sleep time in animals in the pentobarbital-induced sleep model, in contrast to the pretreatment of caffeine [14,15]. Additionally, supplementation with a high dose of theacrine did not affect physiological or mental dysfunction over eight weeks in a clinical trial [16]. Theacrine has been shown to possess several biological activities, including locomotor activation, anti-depression, and anti-metastatic potential on breast cancer cells [14,15,17,18,19,20]. In this study, we speculated that theacrine might serve as an antagonist of the adenosine 2A receptor in a manner similar to caffeine, and that the potential usage of theacrine on breast cancer might also be a result of the blocking of adenosine 2A receptor activation. Firstly, molecular modeling of theacrine docking to the adenosine 2A receptor was theoretically evaluated in comparison with caffeine and adenosine. Subsequently, putative signaling pathways induced by theacrine supplement were examined in MCF-7 human breast cells and an animal model of mammary carcinoma.

## 2. Results

### 2.1. Molecular Modeling of Theacrine, Caffeine and Adenosine Docking to the Adenosine 2A Receptor

To evaluate whether theacrine might serve as an antagonist of the adenosine 2A receptor, the theacrine, in comparison with caffeine (known as an antagonist) and adenosine, was simulated in silico for docking to ADORA2A. The results showed that theacrine as well as caffeine and adenosine could get into and bind to the cavity of ADORA2A (Figure 1). Theacrine formed two conventional hydrogen bonds with Asn253 and Tyr271, and a non-classical hydrogen bond (carbon hydrogen bond) with Glu169 in the binding pocket of ADORA2A. In contrast, caffeine formed one conventional hydrogen bond with Asn253 and a non-classical hydrogen bond with the ADORA2A Glu169. Adenosine formed four conventional hydrogen bonds with the ADORA2A Glu169, Asn253, Ser277, and His278. The same Pi–Pi stacking interaction was observed between the two-ring configuration of theacrine, caffeine or adenosine and the ADORA2A Phe168. Similar alkyl bonds were formed between theacrine or caffeine and the ADORA2A Met177, Leu249, Met270, and Ile274. In contrast, alkyl bonds were formed between adenosine and the ADORA2A Leu249 and Ile274f. Furthermore, dock-fit modeling was employed to estimate the relative strength of theacrine, caffeine and adenosine when interacting with ADORA2A, and the chemical energy was calculated via Discovery Studio software. The results indicated that the binding energy between theacrine and ADORA2A was stronger than that between caffeine and ADORA2A, but slightly weaker than that between adenosine and ADORA2A (Table 1). Among the three compounds, adenosine possessed the strongest binding to ADORA2A, possibly due to its formation of four hydrogen bonds within the binding pocket. The reason why theacrine displayed stronger binding to ADORA2A than caffeine was presumably due to an additional hydrogen bond between the theacrine and the ADORA2A TYR271 (Figure 1). Taken together, the theoretical simulation suggests that theacrine might be an antagonist of the adenosine 2A receptor equivalent to or more effective than caffeine.

### 2.2. Detecting Theacrine as an Antagonist of the Adenosine 2A Receptor

No cytotoxicity was observed when MCF-7 cells were supplemented with theacrine (25, 50, 100, or 200 µM) and/or a specific agonist of the adenosine 2A receptor, CGS21680 (0.25, 0.5, 1, or 2 µM) (Figure 2). Therefore, 100 µM theacrine and 1 µM CGS21680 were used in the following experiments. Adenosine mediates its action via the adenosine 2A receptor coupled primarily to the activation of cAMP, which in turn activates AMP-activated protein kinase (AMPK) [21]. To inspect whether theacrine is an effective antagonist of the adenosine 2A receptor, CGS21680 was employed to activate the adenosine 2A receptor by detecting the level of AMPK phosphorylation (P-AMPK/AMPK) in MCF-7 cells with or without the theacrine supplement (Figure 3). The results showed that the elevated P-AMPK/AMPK ratio by CGS21680 could be significantly reduced when theacrine was concurrently supplemented to MCF-7 cells. Thus, theacrine was demonstrated to be an effective antagonist of the adenosine 2A receptor.

### 2.3. Effects of Theacrine on DMBA-Induced Mammary Carcinoma in Rats

Mammary carcinoma was induced by DMBA in rats with or without the theacrine supplement. Tumor incidence rates were found to be 66.7% (6/9) when rats were treated with DMBA, regardless of the theacrine supplement of 50 or 100 mg/kg body weight (Table 2). Tumor volume was estimated on days 84, 91, 98, and 102, and the average tumor volume in rats treated with DMBA was found to be larger than that in rats treated with DMBA and theacrine (Table 3). Besides body weight, tumors excised from rats were weighed at day 102, and the ratio of tumor over body weight was calculated (Figure 4). It seemed that the development of mammary carcinoma induced by DMBA in rats could be attenuated by a daily supplement of theacrine.

### 2.4. Elevation of Programmed Cell Death in DMBA-Induced Tumor by Theacrine

To see if induction of programmed cell death in tumor cells was involved in the attenuation (tumor volume reduction) of mammary carcinoma by theacrine, expression levels of cleaved-caspase-3/pro-caspase-3 and granzyme B were separately detected. The results showed that the levels of both cleaved-caspase-3/pro-caspase-3 and granzyme B in tumor tissues were significantly elevated when f 50 or 100 mg/kg body weight of theacrine was supplemented daily to rats after DMBA treatment (Figure 5). A relatively high dosage of theacrine (100 mg/kg) seemed to produce better effects than a lower dosage of theacrine (50 mg/kg) on the elevation of both expression levels in tumor tissues.

### 2.5. Effects of Theacrine on the Levels of IFN-γ and TNF-α in DMBA-Treated Rats

Caffeine, serving as an antagonist of the adenosine 2A receptor, has been shown to elevate levels of IFN-γ and TNF-α in tumor cells for the attenuation of cancer development rate [11]. Similarly, significant elevation in IFN-γ level was observed in the serum as well as in the DMBA-induced tumor tissues of rats after the supplement of theacrine (Figure 6A,B). A relatively high dosage of theacrine (100 mg/kg) seemed to produce better effects than a lower dosage of theacrine (50 mg/kg) on the elevation of the IFN-γ level in both examined samples. In contrast, significant elevation in TNF-α level was observed in the serum as well as in the DMBA-induced tumor tissues of rats after the supplementation of a low dosage of theacrine (Figure 6C,D). However, no apparent effects on the TNF-α level in both examined samples were observed when DMBA-treated rats were supplemented with a high dosage of theacrine.

## 3. Discussion

Epidemiologic observation indicated a negative correlation between caffeine consumption and incidence of tumors in humans, and further investigation showed that the therapeutic effects of caffeine were presumably achieved by the antagonism of the adenosine 2A receptor [11,22]. In this study, theacrine was successfully demonstrated to be an antagonist of the adenosine 2A receptor in MCF-7 human breast cells (Figure 3), and the daily oral supplementation of theacrine was shown to produce effective attenuation on the development of mammary carcinoma in an animal model using DMBA-treated rats (Figure 4). The molecular mechanism of the mammary carcinoma attenuation initiated by theacrine was found to be similar to that of the anti-tumor immune response initiated by caffeine, as the induction of programmed cell death as well as elevation of IFN-γ and TNF-α expression levels for anti-tumor immunity was observed in the tumor cells of DMBA-treated rats after daily supplementation of theacrine (Figure 5 and Figure 6). It is suggested that theacrine is an effective antagonist of the adenosine 2A receptor and possesses great potential to be used to attenuate breast cancer.

Caffeine is a purine alkaloid frequently used as a natural stimulant for ergogenic performance all over the world [23]. In addition to the well-known caffeine, some other purine alkaloids, such as theobromine, theophylline and theacrine, have been identified in various tea varieties [24]. The chemical structure of theacrine and caffeine are quite similar, since theacrine is putatively biosynthesized from caffeine via oxidation at the C8 and methylation at the N9 positions [25]. In contrast with the stimulatory effect of caffeine, theacrine has been proposed to possess hypnotic activity [14,17,18]. Due to the side effect of caffeine on sleep quality, theacrine is recommended as a superior solution for an energy boost. In view of the same concern for sleep quality in breast cancer patients, we suggest that theacrine may be a candidate drug which is better than caffeine for the attenuation of the development of mammary carcinoma.

As expected, significant elevation in IFN-γ level was observed in the serum and tumor tissues of the DMBA-induced rats after daily theacrine supplement of 50 or 100 mg/kg (Figure 6A,B). Surprisingly, significant elevation of TNF-α level was observed in the serum and tumor tissues of the DMBA-induced rats after daily supplement of a low dosage (50 mg/kg), but not high dosage (100 mg/kg), of theacrine (Figure 6C,D). It has been reported that the anti-tumor effects of the adenosine 2A receptor blockade, by activating cytotoxic T cells, might be overcome in some tumor microenvironments, and it has been suggested that the effective dosage of adenosine 2A receptor antagonist for cancer immunotherapy might need to be cautiously optimized in order to prevent activation-induced T cell death in tumors [26]. Obviously, the dosage effect and long-term utilization impact of theacrine on the development of mammary carcinoma should be further investigated in more details, such that an optimal dosage of theacrine and its effective duration can be established in order to develop an adequate supporting therapy for breast cancer treatment.

## 4. Materials and Methods

### 4.1. Materials

Theacrine was purchased from Bolise (Shanghai, China). A specific adenosine 2A subtype receptor agonist, CGS21680, was purchased from Cayman Chemical (Ann Arbor, MI, USA). The 7,12-Dimethylbenz[a]anthracene (DMBA) was purchased from Sigma-Aldrich (St. Louis, MO, USA). Bio-Rad Protein Assay Dye Reagent concentrate, ClarityTM and Clarity MaxTM Western ECL Blotting Substrate were bought from Bio-Rad Laboratories (Hercules, CA, USA). Protease inhibitor cocktail and phosphatase inhibitor cocktail were purchased from Roche Molecular Systems (Pleasanton, CA, USA). Mouse Anti-Tubulin Antibody was bought from Thermo Fisher Scientific. Anti-Granzyme B antibody (ab53097), P-AMPK antibody and AMPK antibody were bought from Abcam (Cambridge, UK). Caspase-3 antibody, GAPDH antibody, goat anti-mouse IgG antibody and goat anti-rabbit IgG antibody were bought from Cell Signaling Technologies (Danvers, MA, USA).

### 4.2. Homology Modeling and Docking

The detailed homology modeling of ADORA2A as the adenosine 2A receptor for ligand docking was described in a previous study [27]. The 3D structures of the adenosine 2A receptor (Protein Data Bank: 3EML) was obtained from the PDB database, and the 3D structures of theacrine, caffeine and adenosine were obtained from the ChemSpider database. The modeling ADORA2A structure was used for ligand docking of theacrine, caffeine (antagonist of ADORA2A), and adenosine (agonist of ADORA2A) through the Dock Ligands (Libdock) module of the Discovery Studio software. The binding pocket was defined by the active center provided by the Protein Data Bank.

### 4.3. Cell Culture

The MCF-7 human breast cell line was cultured in Minimum Essential Media (MEM) supplemented with 10% FBS, 1 mM sodium pyruvate, 1.5 g/L sodium bicarbonate, and 1% penicillin and streptomycin, and kept in an incubator with 5% CO_2_ at 37 °C. The MCF-7 cells were grown in 6-well flat-bottomed culture plates at a concentration of 5 × 10^5^ per well in 2 mL of complete culture medium.

### 4.4. Cell Viability Assay

Cell viability was measured by 3-(4,5-Dimethylthiazol-2-yl)-2,5-diphenyltetrazolium bromide (MTT) assay (Biological Industries, Cromwell, CT, USA). The MCF-7 cells were loaded in 96-well flat-bottomed culture plates at a concentration of 2 × 10^4^ cells/mL. They were then treated with or without theacrine (25, 50, 100 and 200 µM) and CGS21680 (0.25, 0.5, 1 and 2 μM), as used in previous studies [19,28]. After treated for 24 h, an MTT solution was added to the wells and incubated at 37 °C for 4 h according to the manufacturer’s instructions. Absorbance was measured at 540 nm with a Multiskan GO Microplate Spectrophotometer (Thermo Fisher Scientific, Waltham, MA, USA).

### 4.5. Animals

Female Sprague–Dawley rats (170–190 g, 50-day-old) bought from BioLasco, Taiwan Co., Ltd. (Taipei, Taiwan) were kept in the animal house of the Department of Veterinary Medicine, National Chung Hsing University under controlled environments of 23 ± 2 °C, 60 ± 10% humidity and 12-h light/dark cycle. Rats were fed with a standard chow diet (calories provided by approximately 30% protein, 13.5% fat, and 56.5% carbohydrate, 5001 Rodent Lab Diet, St. Louis, MO, USA). They were allowed free access to chow diet and tap water. The rats were adapted for 1 week, and then divided into four groups comprising of (i) Control group: rats were allowed free access to tap water (*n* = 6), (ii) DMBA group: rats were administered with 100 mg/kg of DMBA for tumor induction and allowed free access to tap water (*n* = 9), (iii) Low-dose theacrine (LD-TC) group: rats were administered with 100 mg/kg of DMBA for tumor induction and allowed free access to tap water supplemented with 50 mg/kg BW/day of theacrine (*n* = 9), and (iv) High-dose theacrine (HD-TC) group: rats were administered with 100 mg/kg of DMBA for tumor induction and allowed free access to tap water supplemented with 100 mg/kg BW/day of theacrine (*n* = 9). To ensure that the rats could receive the designed dose of theacrine, they were kept in an individual cage with the bottle containing theacrine or water (15 mL) in the morning (8 a.m.) and returned to their origin cage (two rats in a house) in the afternoon (6 a.m.). In 102 days post-DMBA exposure, all animals were sacrificed. The animal operation was permitted by the Institutional Animal Care and Use Committee of the the National Chung-Hsing University with approval number IACUC 109-077.

### 4.6. Estimation of Body and Tumor Parameters

Weights of the rats were individually recorded during the period of experiment. The incidence of tumor was examined and its volume was estimated using Π × (a) × (b)^2^/2, where ‘a’ is the shortest and ‘b’ is the longest length of the tumor. Tumor number was the number of tumors observed in the experimental rats. Tumor incidence was defined as the fraction of tumors detected in each group.

### 4.7. Enzyme Linked Immune Sorbent Assay for Proinflammatory Cytokines

Quantification of interferon-γ (IFN-γ) or tumor necrosis factor-α (TNF-α) was carried out by enzyme linked immunosorbent assay (ELISA). Homogenates of mammary tissue samples were subjected to centrifugation at 12,000× *g* for 15 min, and the supernatants were collected for further analyses. The levels of IFN-γ and TNF-α were estimated in the supernatants using specific rat ELISA kits. Rat IFN-γ Quantikine ELISA Kit and Rat TNF-α Quantikine ELISA Kit were bought from R&D System (Minneapolis, NE, USA).

### 4.8. Western Blotting

After anesthesia, the tumors were dissected and extracted with T-PER tissue protein extraction buffer (Pierce Biotechnology, Thermo-Fisher, Rockford, IL, USA) supplemented with an inhibitor cocktail of protease and phosphatase (Calbiochem, Merck Millipore, Darmstadt, Germany). After various treatments, MCF-7 cells were lysed in the M-PER reagent (Thermo-Fisher Scientific, Waltham, MA, USA). The supernatants were collected after centrifugation at 15,000× *g* for 10 min at 4 °C and used for the following analyses. After separated by SDS-PAGE, proteins extracted from various samples were transferred to polyvinylidene difluoride membranes (Merck Millipore). The membranes were immersed in 5% skim milk and then allowed to be recognized by antibodies against tubulin (1:5000, GeneTex, Inc., San Antonio, TX, USA), caspase-3 (1:1000; Cell Signalling Technology, Inc., Beverly, MA, USA), and GAPDH (1:5000; Merck Millipore). Followed by further recognition with HRP-conjugated secondary antibody (a 1:5000 dilute of anti-rabbit or anti-mouse IgG), the target protein bands were revealed with Immobilon™ Western Chemiluminescent HRP Substrate reagent (Merck Millipore) and quantified by chemiluminescence with MiniChemi I system (Beijing Sage Creation Science, Beijing, China). Relative protein content was shown as folds of the protein content in the control group after being normalized with the contents of tubulin and GAPDH.

### 4.9. Statistical Analysis

Data were shown as mean ± standard deviation (S.D.). One-way analysis of variance (ANOVA) and post-hoc analysis of Tuckey’s test were used to evaluate significant differences between various groups. Statistical calculation was performed by SigmaStat (version 12.0) with *p* < 0.05 defined to be statistically significant.

## Figures and Tables

**Figure 1 molecules-26-07455-f001:**
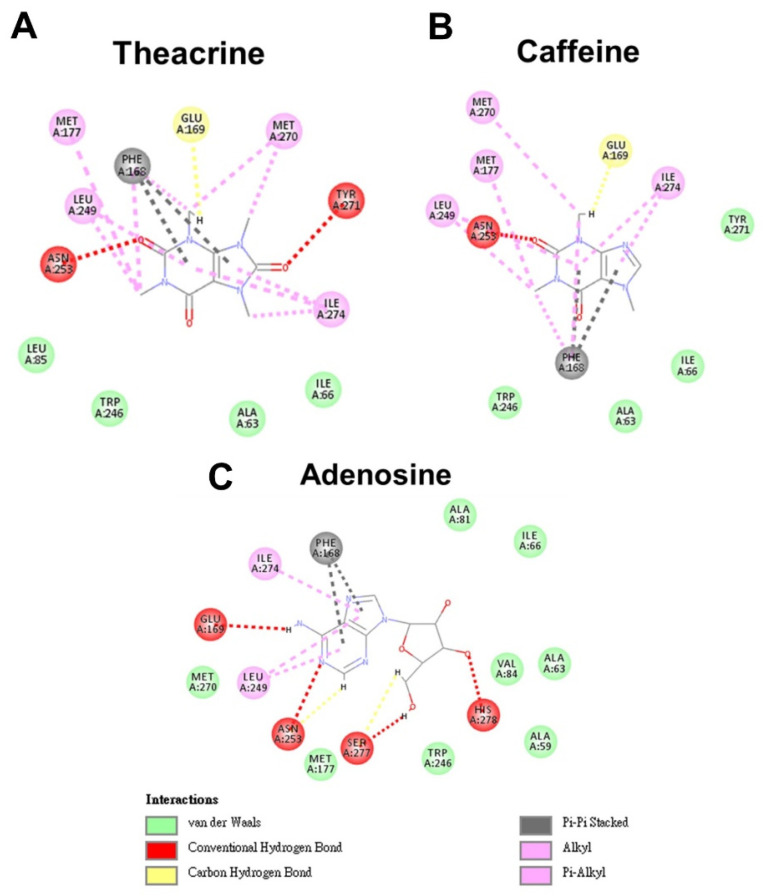
Molecular modeling of intermolecular interaction between the binding pocket of the adenosine 2A receptor (ADORA2A) and theacrine (**A**), caffeine (**B**) or adenosine (**C**). The red dashed lines are conventional hydrogen bonds. The yellow dashed lines are carbon hydrogen bonds. The green color represents Van der Waals force. The gray dashed lines show Pi–Pi stacking. The pink dashed lines indicate alkyl bonds. The amino acids of ADORA2A involved in the interaction with the three ligands are shown in colored balls.

**Figure 2 molecules-26-07455-f002:**
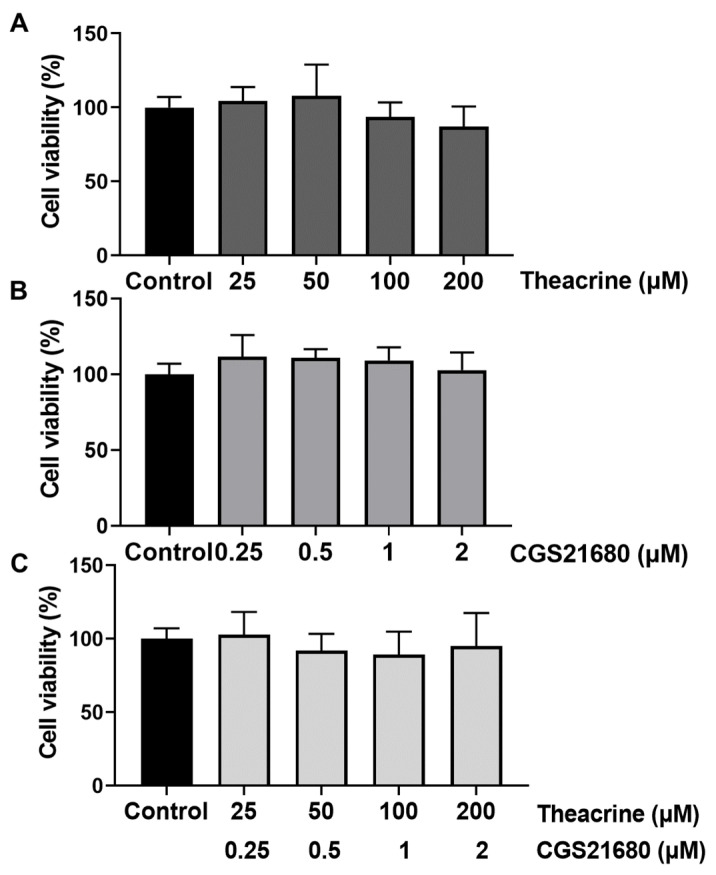
Effects of theacrine and CGS21680 on cell viability in MCF-7 cells. MCF-7 cells were treated with theacrine of 25, 50, 100, and 200 µM (**A**) and/or with CGS21680 of 0.25, 0.5, 1, and 2 µM for 24 h (**B**,**C**). Cell viability was calculated by the MTT assay. The results were shown as mean ± SD (*n* = 6).

**Figure 3 molecules-26-07455-f003:**
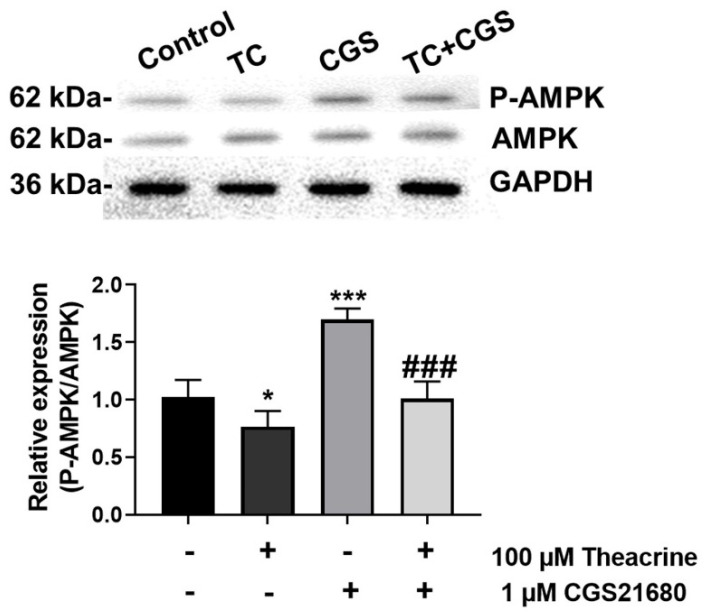
Effects of theacrine on the relative expression of P-AMPK/AMPK induced by CGS21680 in MCF-7 cells. MCF-7 cells were treated with theacrine (100 µM) and CGS21680 (1 µM) for 24 h. Cells were harvested and analyzed through western blotting using antibodies against P-AMPK and AMPK. The expression of GAPDH was used as the internal control. All data were expressed as mean ± SD (*n* = 4). Values were analyzed by one-way ANOVA followed by Tukey post hoc test, with statistically significant at *p* < 0.05; * *p* < 0.05, *** *p* < 0.001 as compared with the control group, and ### *p* < 0.001 as compared with CGS21680 alone group.

**Figure 4 molecules-26-07455-f004:**
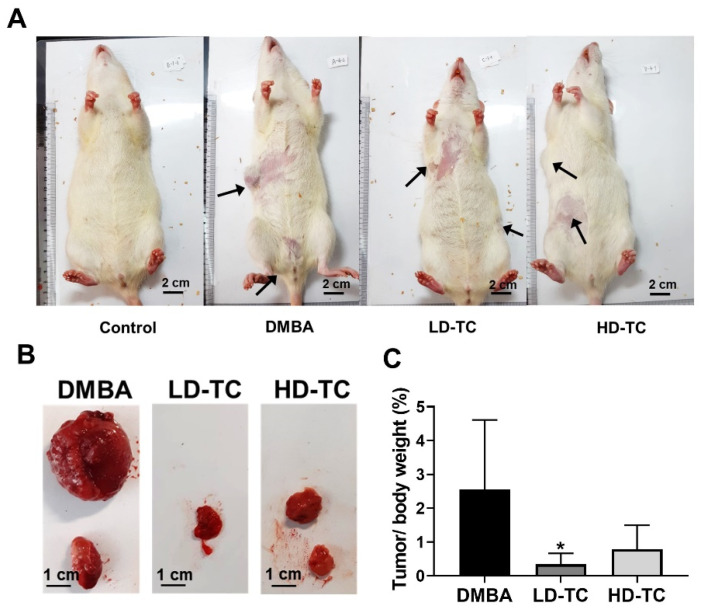
Effects of theacrine on the DMBA-induced mammary carcinoma rat model. Female Sprague–Dawley rats were administered with a single dose of DMBA (100 mg/kg) or vehicle through intra gastric route, and then daily administered either water or theacrine (50 or 100 mg/kg in drinking water) for 102 days. (**A**) The images showed the mammary adenocarcinoma in DMBA-induced rats. (**B**) Representative tumor images. (**C**) Effects of theacrine on the ratio of tumor over body weight. Value was found statistically significant at * *p* < 0.05 as compared with the DMBA group.

**Figure 5 molecules-26-07455-f005:**
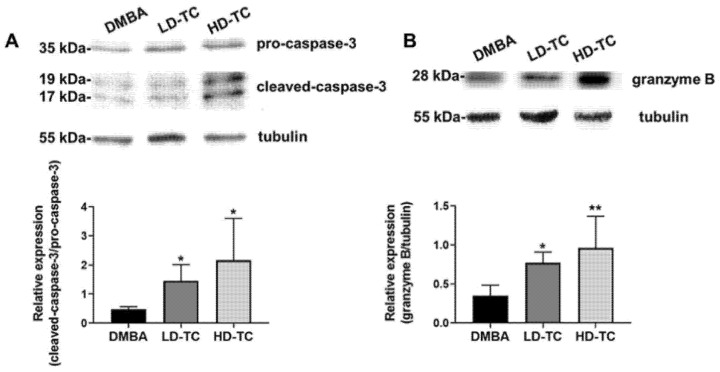
Effects of theacrine on the expression levels of cleaved-caspase-3/pro-caspase-3 and granzyme B in tumor tissues of DMBA-treated rats. Tumor tissues were harvested and analyzed through western blotting using antibodies against caspase-3 (**A**) and granzyme B (**B**). The expression of tubulin was used as an internal control. The results were shown as mean ± SD (*n* = 6). Values were found statistically significant at *p* < 0.05; * *p* < 0.05, ** *p* < 0.01 as compared with the DMBA group.

**Figure 6 molecules-26-07455-f006:**
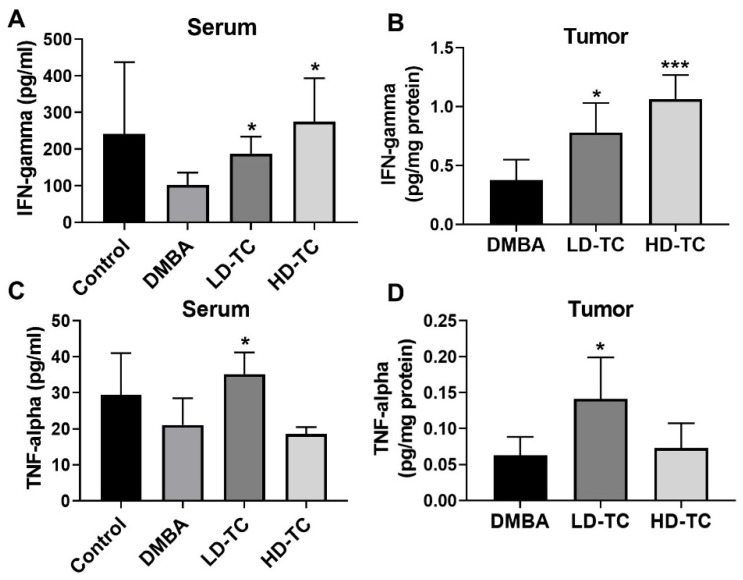
Effects of theacrine on the IFN-γ and TNF-α levels in serum and tumor tissue of DMBA-treated rats. Levels of IFN-γ in serum (**A**) and tumor tissue (**B**) as well as TNF-α in serum (**C**) and tumor tissue (**D**) were quantitated by ELISA. The results were shown as mean ± SD (*n* = 5 for serum sample and *n* = 6 for tumor tissue). Values were found statistically significant at *p* < 0.05; * *p* < 0.05, *** *p* < 0.001 as compared with the DMBA group.

**Table 1 molecules-26-07455-t001:** Theoretic binding energy for the interaction between the binding pocket of the adenosine 2A receptor (ADORA2A) and theacrine, caffeine or adenosine.

Ligand	Binding Energy (kJ/mol)	Absolute Energy (kJ/mol)	Complex Energy (kJ/mol)
Theacrine	−105.892	32.5403	−18,472.1
Caffeine	−101.077	27.6803	−18,479.5
Adenosine	−106.911	61.2889	−18,555.9

**Table 2 molecules-26-07455-t002:** Effects of theacrine on tumor incidence and body weight.

Parameter	Control ^x^	DMBA	LD-TC	HD-TC
Tumor incidence (%) ^y^	0.0	66.7	66.7	66.7
Body weight (g)	330.52 ± 22.86	314.51 ± 24.74	315.48 ± 25.19	329.94 ± 20.26

^x^: Control group: Rats were administered with single dose of soybean oil during week ‘0′ (*n* = 6 per group). DMBA group: Rats were administered with single dose of 1 mL DMBA (100 mg/kg) in soybean oil during week ‘0′ (*n* = 9 per group). LD-TC (low-dose theacrine) group: Rats were administered with single dose of 1 mL DMBA (100 mg/kg) in soybean oil during week ‘0′ and were administered daily with 50 mg/kg theacrine in drinking water (*n* = 9 per group). HD-TC (high-dose theacrine) group: Rats were administered with single dose of 1 mL DMBA (100 mg/kg) in soybean oil during week ‘0′ and were administered daily with 100 mg/kg theacrine in drinking water (*n* = 9 per group). ^y^: All data are expressed as mean ± S.D. for each group.

**Table 3 molecules-26-07455-t003:** Effects of theacrine on tumor volume.

		Tumor Volume		
Group	Day 84	Day 91	Day 98	Day 102
Control ^x^	0.00 ^y^	0.00	0.00	0.00
DMBA	1.38 ± 0.76	2.76 ± 1.53	4.13 ± 2.76	11.62 ± 7.60
LD-TC	0.57 ± 0.55	0.71 ± 0.75 *^z^	0.96 ± 1.13 *	3.46 ± 4.77 *
HD-TC	0.87 ± 0.87	1.72 ± 1.86	2.37 ± 2.39	4.71 ± 3.84

^x^: Four groups were the same as described in Table 2. ^y^: The tumor volume was measured using, π/6 × (a) 2 × (b), where ‘a’ is the shortest and ‘b’ is the longest length of the tumor. All data are expressed as mean ± SD for groups of six rats in each. The unit of tumor volume is cm3. ^z^: Values are statistically significant at *p* < 0.05. * *p* < 0.05 as compared with DMBA group (*n* = 6).

## Data Availability

Not applicable.

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
