# Peer review of "Attenuation of Tumor Development in Mammary Carcinoma Rats by Theacrine, an Antagonist of Adenosine 2A Receptor"

_molecules, 2021, doi:10.3390/molecules26247455_

Round 1

Reviewer 1 Report

The main finding of this paper, namely, that tumor growth in an animal model of breast carcinoma can be significantly inhibited by administration of an adenosine A2A receptor antagonist, is definitely interesting and deserves to be published. Based on the results reported in the manuscript, this beneficial action of an A2AR antagonist seems to be due both to an enhancement of the immune system response and the induction of programmed cell death in tumor cells. However, there are some methodological issues with this manuscript and in my opinion, the text would benefit from some re-organization.

MAIN COMMENTS

1. The rationale for the choice of theacrine as a therapeutic drug is not clearly stated in the Introduction, and emerges only in the Discussion. Some information on this subject should be provided in the Introduction, e.g. the fact that theacrine may be expected to have similar actions as caffeine, but has less impact on sleep. The authors could perhaps also mention that theacrine is a natural substance that may be administered in rather high dose (although I'm not sure that this is in fact the case, is anything known about the toxicity of theacrine?). May the hypnotic action of theacrine cause problems if this drug is administered to humans at high doses?

2. The question concerning toxicity is important because a dose of 50 to 100 mg/kg body weight/day seems very high. This would correspond to 3.5 to 7 grams daily for a 70 kg human. How was this dose selected? Are lower doses ineffective?

3. The comparison of 50 and 100 mg/kg body weight/day seems a bit strange, since there is only a 2-fold difference between the two doses. A 10-fold difference (such as 10 and 100 mg/kg body weight/day) would seem a more logical choice.

4. How sure are the authors that the rats really received 50 and 100 mg/kg body weight/day? Since theacrine was administered via the drinking water, and the animals had free access to this water, the actual dose that they received was dependent on the amount of water that they consumed. Particularly at the high dose, the rats may not have liked the taste of the water and may have reduced their water intake. The effect of the high dose of theacrine on several measured parameters was lower than the effect of the low dose. This suggests that the animals drank less if the water contained a higher concentration of theacrine. Was water intake measured in the study? Animals can also spill water, thus exact determination of the water consumption of a rat may be quite difficult. Were animals individually housed? More detail on the dosing procedures should be provided.

5. Why were no competition binding studies with theacrine performed, using a commercial radioligand for A2A receptors? The IC50 values of theacrine should be compared to those of caffeine and a registered A2A receptor antagonist, like istradefylline. Is theacrine more A2A-subtype-selective than caffeine? Caffeine binds not only to A2A but also to A1 receptors. Has theacrine a lower affinity for the A1 subtype than caffeine and therefore less impact on sleep? Are there any literature data on the receptor binding profile of theacrine?

MINOR COMMENTS

6. The abbreviatons LD-TC and HD-TC appear early in the text, but are defined not earlier than in section 4.5. Abbreviations should probably be defined (written in full) at the first occurrence.

7. The English in line 45 is not correct (authors write: "is bitter than caffeine in taste"). Do they mean: "is more bitter than caffeine in taste", or: "has a bitter taste, like caffeine"? Please clarify.

Author Response

Reviewer 1

The main finding of this paper, namely, that tumor growth in an animal model of breast carcinoma can be significantly inhibited by administration of an adenosine A2A receptor antagonist, is definitely interesting and deserves to be published. Based on the results reported in the manuscript, this beneficial action of an A2AR antagonist seems to be due both to an enhancement of the immune system response and the induction of programmed cell death in tumor cells. However, there are some methodological issues with this manuscript and in my opinion, the text would benefit from some re-organization.

MAIN COMMENTS

  1. The rationale for the choice of theacrine as a therapeutic drug is not clearly stated in the Introduction, and emerges only in the Discussion. Some information on this subject should be provided in the Introduction, e.g. the fact that theacrine may be expected to have similar actions as caffeine, but has less impact on sleep. The authors could perhaps also mention that theacrine is a natural substance that may be administered in rather high dose (although I'm not sure that this is in fact the case, is anything known about the toxicity of theacrine?). May the hypnotic action of theacrine cause problems if this drug is administered to humans at high doses?

Response: Thanks for your kind suggestions.

According to the reviewer’s comment, we added some statements in revised manuscript as follow:

Changed revision (red mark):

Line 49-55

“Caffeine and theophylline have been reported to induce anti-tumor immune re-sponse for attenuating breast cancer by blocking adenosine 2A receptor [11, 12]. Theacrine, a purine alkaloid structurally similar to caffeine, is more bitter than caffeine in taste, and found as a relatively abundant constituent in Yunnan Kucha tea [13]. Although theacrine and caffeine possess similar structure backbone, they display quite different physiological functions in central nervous system. In a previous study, pretreat-ment of theacrine prolonged the sleep time in the pentobarbital-induced sleep animal model in contrast to the pretreatment of caffeine [14]. Additionally, supplemention with high dose of theacrine did not affect physioloical or mental dysfunction over 8 weeks in a clinical trial [15]. Theacrine has been shown to possess several biological activities, including locomotor activation, anti-depression, and anti-metastatic potential on breast cancer cells [14, 16-20].

  • The question concerning toxicity is important because a dose of 50 to 100 mg/kg body weight/day seems very high. This would correspond to 3.5 to 7 grams daily for a 70 kg human. How was this dose selected? Are lower doses ineffective?

Response:

To estimate the anti-tumor activity of theacrine, we referred to the dose of caffeine in the previous study, related to the tumor animal model [1]. The study used 0.1% caffeine in water, approximately equal to 1000 mg/kg in mice. Accroding to the average body weight and surface area,  the dose of the mouse is equal to two times the dose in the rat [2]. Thus, we suggested that the dose of theacrine in rats could be 500 mg/kg/day. Thus, we selected 50mg/kg as the low dose. Based on the LD50 of caffeine orally to rats being 367mg/kg [3], we selected 50mg/kg as the low dose and 100mg/kg as the high dose.

The toxicity of high doses will be a concern. As shown in a previous study, the LD50 of caffeine orally to rats is 367mg/kg [3]. In light of the safety consideration, we selected 100mg/kg of theacrine as a  high dose in the study.

Reference:

  1. Venkata Charan Tej GN, Neogi K, Verma SS, Chandra Gupta S, Nayak PK. Caffeine-enhanced anti-tumor immune response through decreased expression of PD1 on infiltrated cytotoxic T lymphocytes. Eur J Pharmacol. 2019 Sep 15;859:172538. doi: 10.1016/j.ejphar.2019.172538. Epub 2019 Jul 13. PMID: 31310752.
  2. Nair AB, Jacob S. A simple practice guide for dose conversion between animals and human. J Basic Clin Pharm. 2016 Mar;7(2):27-31. doi: 10.4103/0976-0105.177703. PMID: 27057123; PMCID: PMC4804402.
  3. Adamson RH. The acute lethal dose 50 (LD50) of caffeine in albino rats. Regul Toxicol Pharmacol. 2016 Oct;80:274-6. doi: 10.1016/j.yrtph.2016.07.011. Epub 2016 Jul 25. PMID: 27461039.

  • The comparison of 50 and 100 mg/kg body weight/day seems a bit strange, since there is only a 2-fold difference between the two doses. A 10-fold difference (such as 10 and 100 mg/kg body weight/day) would seem a more logical choice.

Response: 

The reviewer is right, we originally thought to use 10 and 100 mg/kg in this study, but we were afraid of the low dose of 10 mg/kg might not result in a significant effect.  Therefore, we decided to design in a more conservative way by using  50 and 100 mg/kg.  We are not sure whether 10 mg/kg of theacrince is effective or not at this moment.

  • How sure are the authors that the rats really received 50 and 100 mg/kg body weight/day? Since theacrine was administered via the drinking water, and the animals had free access to this water, the actual dose that they received was dependent on the amount of water that they consumed. Particularly at the high dose, the rats may not have liked the taste of the water and may have reduced their water intake. The effect of the high dose of theacrine on several measured parameters was lower than the effect of the low dose. This suggests that the animals drank less if the water contained a higher concentration of theacrine. Was water intake measured in the study? Animals can also spill water, thus exact determination of the water consumption of a rat may be quite difficult. Were animals individually housed? More detail on the dosing procedures should be provided.

Response:. 

We added some animal experiment procedure to the revised manuscript as follow:

Changed revision:

Line 287-290

 “… To ensure that rats could receive the designed dose of theacrine, they were kept in an individual cage with the bottle containing theacrine or water (15 mL) in the morning (8 a.m.) and returned to their origin cage (two rats in a house) in the afternoon (6 a.m.). ….”

  • Why were no competition binding studies with theacrine performed, using a commercial radioligand for A2A receptors? The IC50 values of theacrine should be compared to those of caffeine and a registered A2A receptor antagonist, like istradefylline. Is theacrine more A2A-subtype-selective than caffeine? Caffeine binds not only to A2A but also to A1 receptors. Has theacrine a lower affinity for the A1 subtype than caffeine and therefore less impact on sleep? Are there any literature data on the receptor binding profile of theacrine?

Response: thanks for your kind comments. 

In the study, we didn’t investigate the relationship between theacrine and A1A receptors. As the reviewer’s suggestion using a commercial radioligand for A2A receptors is a good way to clarify the relationship between theacrine and A2A or theacrine and A1A  receptors.  But, the development of specific radioligands or radioreceptor assays is not allowed at our laboratory. In the following experiments, we could find a radioactivity-allowed laboratory to do competition binding studies. In this work, we mainly focused on the effects of theacrine on breast carcinoma.

MINOR COMMENTS

  1. The abbreviatons LD-TC and HD-TC appear early in the text, but are defined not earlier than in section 4.5. Abbreviations should probably be defined (written in full) at the first occurrence.

Response: thanks for the correction.

Changes are highlighted in red as follows:

Line 141-144

“x: Control group: Rats were administered with single dose of soybean oil during week ‘0’ (n = 6 per group). DMBA group: Rats were administered with single dose of 1 ml DMBA (100 mg/kg) in soybean oil during week ‘0’ (n = 9 per group). LD-TC (low-dose theacrine) group: Rats were ad-ministered with single dose of 1 ml DMBA (100 mg/kg) in soybean oil during week ‘0’, and were administered daily with 50 mg/kg theacrine in drinking water (n = 9 per group). HD-TC (high-dose theacrine) group: Rats were administered with single dose of 1 ml DMBA (100 mg/kg) in soybean oil during week ‘0’, and were administered daily with 100 mg/kg theacrine in drinking water (n = 9 per group). y: All data are expressed as mean ± S.D. for each group.”

  1. The English in line 45 is not correct (authors write: "is bitter than caffeine in taste"). Do they mean: "is more bitter than caffeine in taste", or: "has a bitter taste, like caffeine"? Please clarify.

Response: thanks for the correction.

Line 46

Theacrine, a purine alkaloid structurally similar to caffeine, is more bitter than caffeine in taste, and found as a relatively abundant constituent in Yunnan Kucha tea [13].

Reviewer 2 Report

The paper by Jhuo et al. is interesting and well written, it reports that theacrine is an antagonist of the A2A adenosine receptor in MCF-7 human breast cells. Moreover, it shows that daily oral supplement of theacrine was shown to possess effective attenuation on the development of mammary carcinoma in an animal model using DMBA-treated rats.

The paper is worthy of publication in Molecules but I think that some minor revisions are required:

- The author should enumerate the references as reported in the main text;

- The authors should consider reporting the antagonist behaviuor of theacrine towards A2A adenosine receptors also through a cAMP assay to corroborate P-AMPK/AMPK data;

- In the discussion, the reason why there is a significant increase of TNF-α level in the serum and tumor tissues of the DMBA-induced rats after daily supplement of low dosage 218 (50 mg/kg), but not high dosage (100 mg/kg) of theacrine should be deeper discussed;

Author Response

Reviewer 2

The paper by Jhuo et al. is interesting and well written, it reports that theacrine is an antagonist of the A2A adenosine receptor in MCF-7 human breast cells. Moreover, it shows that daily oral supplement of theacrine was shown to possess effective attenuation on the development of mammary carcinoma in an animal model using DMBA-treated rats.

The paper is worthy of publication in Molecules but I think that some minor revisions are required:

- The author should enumerate the references as reported in the main text;

Response: thanks for your kind suggestions.

Following reviewer’s suggestions, we added references in the revised manuscript.

- The authors should consider reporting the antagonist behaviuor of theacrine towards A2A adenosine receptors also through a cAMP assay to corroborate P-AMPK/AMPK data;

Response: Thanks for your kind suggestions.  

According to previous studies [1-2], activation of A2A receptor will increase the level of AMPK phosphorylation. Therefore, we evaluated AMPK phosphorylation as activative marker of A2A receptor. To confirm whether theacrine could blockade adenosine 2 receptor cell signaling, we detected the level of P-AMPK /AMPK.

Reference:

  1. Pang T, Rajapurohitam V, Cook MA, Karmazyn M. Differential AMPK phosphorylation sites associated with phenylephrine vs. antihypertrophic effects of adenosine agonists in neonatal rat ventricular myocytes. Am J Physiol Heart Circ Physiol. 2010 May;298(5):H1382-90. doi: 10.1152/ajpheart.00424.2009. Epub 2010 Feb 26. PMID: 20190100.
  2. Li YF, Ouyang SH, Tu LF, Wang X, Yuan WL, Wang GE, Wu YP, Duan WJ, Yu HM, Fang ZZ, Kurihara H, Zhang Y, He RR. Caffeine Protects Skin from Oxidative Stress-Induced Senescence through the Activation of Autophagy. Theranostics. 2018 Nov 10;8(20):5713-5730. doi: 10.7150/thno.28778. PMID: 30555576; PMCID: PMC6276298.

- In the discussion, the reason why there is a significant increase of TNF-α level in the serum and tumor tissues of the DMBA-induced rats after daily supplement of low dosage 218 (50 mg/kg), but not high dosage (100 mg/kg) of theacrine should be deeper discussed

Response: Thanks for your kind comments.

According to Figure 6, we suggested the anti-cancer activity of theacrine majorly through the increase of IFN-γ levels. As the reviewer’s comments, only a high dose of the theacrine had a significantly decrease of TNF-alpha in the DMBA-induced rats in contrast to the low dose of the theacrine. A previous study showed anti-inflammation activity of theacrine through decreased TNF-α levels [1]. Therefore, it might be one of the reasons why the high dose of the theacrine did not result in a significant increase of TNF-α level. Of course, the explanation should be further investigated in the future.

Reference:

  1. Li WX, Li YF, Zhai YJ, Chen WM, Kurihara H, He RR. Theacrine, a purine alkaloid obtained from Camellia assamica var. kucha, attenuates restraint stress-provoked liver damage in mice. J Agric Food Chem. 2013 Jul 3;61(26):6328-35. doi: 10.1021/jf400982c. Epub 2013 Jun 18. PMID: 23678853.
